# A Global Perspective of Racial–Ethnic Inequities in Dental Caries: Protocol of Systematic Review

**DOI:** 10.3390/ijerph19031390

**Published:** 2022-01-26

**Authors:** Sonia Nath, Sneha Sethi, João L. Bastos, Helena M. Constante, Kostas Kapellas, Dandara Haag, Lisa M. Jamieson

**Affiliations:** 1Australian Research Centre for Population Oral Health, Adelaide Dental School, University of Adelaide, Adelaide 5000, Australia; sneha.sethi@adelaide.edu.au (S.S.); kostas.kapellas@adelaide.edu.au (K.K.); dandara.haag@adelaide.edu.au (D.H.); lisa.jamieson@adelaide.edu.au (L.M.J.); 2Graduate Program in Public Health, Federal University of Santa Catarina, Florianópolis 88040-900, SC, Brazil; joao.luiz.epi@gmail.com (J.L.B.); lenaconstante@gmail.com (H.M.C.)

**Keywords:** cultural, dental caries, ethnicity, inequalities, minority, oral health, race

## Abstract

Though current evidence suggests that racial–ethnic inequities in dental caries persist over time and across space, their magnitude is currently unknown from a global perspective. This systematic review aims to quantify the magnitude of racial/ethnic inequities in dental caries and to deconstruct the different taxonomies/concepts/methods used for racial/ethnic categorization across different populations/nations. This review has been registered in PROSPERO; CRD42021282771. An electronic search of all relevant databases will be conducted until December 2021 for both published and unpublished literature. Studies will be eligible if they include data on the prevalence or severity of dental caries assessed by the decayed, missing, filled teeth index (DMFT), according to indicators of race-ethnicity. A narrative synthesis of included studies and a random-effects meta-analysis will be conducted. Forest plots will be constructed to assess the difference in effect size for the occurrence of dental caries. Study quality will be determined via the Newcastle–Ottawa Scale and the GRADE approach will be used for assessing the quality of evidence. This systematic review will enhance knowledge of the magnitude of racial/ethnic inequities in dental caries globally by providing important benchmark data on which to base interventions to mitigate the problem and to visualize the effects of racism on oral health.

## 1. Introduction

Over the last four decades, there have been heated debates about the drivers of general and oral health inequities [1,2,3]. The focus of public health research has moved from individual-level approaches to the health impacts of broader social forces [4]. The framework of structural inequities, for instance, is used to refer to the social, environmental, economic, and cultural determinants of health, which often revolve around the role of race, ethnicity, gender, employment, disability, immigration status, and place in the emergence of health inequities between and within nations [2,5]. This framework thus contends that health inequities stem from unfair differences in access to power and resources among countries and their constituent groups [5].

Every country has a different way of categorizing ethnic, racial, and cultural groups, with no global standard definition [6,7]. Oral health researchers have been challenged with documenting these complex concepts without a standard definition [8]. Thus, interpretation of results becomes difficult as these classifications can either over- or underestimate outcome measures. Another concern is the comparison and integration of results across studies become complicated and meaningless. Use of terms such as “race” and “ethnicity” are often blurred and used interchangeably, with the features of racial grouping not distinct from an ethnic one [9]. Historically, race was considered a biological concept (similar physical trait), but more recent definitions take it as a marker of varied and intersecting social processes, including phenotype expression, interpersonal discrimination, and residential segregation, to name a few (IOM, 2009, [10]). The term “race” is fundamentally flawed and is an arbitrary classification of modern humans still used in the fields of science and medicine [11]. “Ethnicity” refers to a cluster of people who have common cultural traits, such as a common language, geographic location, religion, traditions, values, beliefs, and a sense of history. Similarly to race, ethnicity is not fixed; it is self-defined, open, flexible, and subject to change [11,12].

Racial–ethnic health inequities refer to unfair differences in the health outcomes between groups based on race, ethnicity, or any other markers of ancestral or cultural heritage [13]. Several researchers have documented over the last three decades the disparities in the quality of healthcare among racial and ethnic minorities and non-minorities. Racial–ethnic health inequities have been documented for a range of health outcomes, including higher mortality, the recent COVID-19 pandemic [14], psychiatric disorders, cardiovascular disease, cancer, HIV/AIDS, diabetes, and other chronic and infectious diseases [15,16]. A higher prevalence of untreated tooth decay (caries), severe periodontal disease, and complete or partial edentulism have been observed among racially marginalized groups [17,18,19,20]. The prevalence of dental caries has been disproportionately high among minoritized racial and ethnic groups, despite pronounced changes in related risk factors, treatment, and access to treatment of oral diseases [18,20]. Untreated dental caries can affect the overall quality of life by interfering with speech, eating, sleep, and work by causing pain. Cost can be a barrier for treating dental caries and the minority groups are often faced with financial burdens and have difficulties affording and accessing oral health care services [21].

To the best of our knowledge, no systematic review has estimated the disproportionally higher burden of dental caries among racially marginalized groups globally. Current reviews have restricted their search strategy to Indigenous populations [20,22,23,24], or other groups, [25,26], often without any comparator. Systematic reviews and meta-analyses provide the highest quality evidence from a clinical, research, or policy perspective. A preliminary search of appropriate sources including PROSPERO, MEDLINE, and the Cochrane Database of Systematic Reviews, revealed no published or ongoing systematic reviews addressing the following research question: to determine the magnitude of racial–ethnic inequities in the prevalence and severity of dental caries worldwide. This paper will additionally describe the theory and methods used to date to classify ethnic and racial groups in dental caries research. We hypothesize that the methods of classifying population groups have a direct impact on the ways health disparities are examined and interpreted.

The research question has been defined according to the PECOS format; population (P), exposure (E), comparator (C), outcomes (O), study design (S):To quantify the magnitude of inequities in dental caries prevalence and severity (O) among racial–ethnic (E) groups (P) when compared to the non-minority (C) in observational studies (S). What concepts and methods are used by researchers to address race, ethnicity, or combination of both for a group of the population? Are the concepts of race and ethnicity treated differently by the authors?

## 2. Materials and Methods

This systematic review and meta-analysis will follow the Preferred Reporting Items for Systematic Reviews and Meta-Analysis Protocol (PRISMA-P) guidelines [27] (Appendix A). This review has been registered in PROSPERO; CRD42021282771.

### 2.1. Eligibility Criteria

#### 2.1.1. Participants

For this review, we aim to estimate dental caries prevalence and severity among both racially marginalized and hegemonic groups within a country. Participants of all ages, including children and adults of all genders, will be eligible for inclusion. The review will not be restricted to any geographic location.

#### 2.1.2. Exposure

Exposure is defined as members of an ethnically or racially minoritized group in a given country. The racial–ethnic definitions or classification systems vary across countries and populations, meaning that a single global definition is challenging. We intend to identify the different taxonomic classifications used in dental caries literature for racial/ethnic categories (e.g., U.S. Office of Management and Budget, based in the country of birth, religion, et cetera) or identify other concepts used for classification of groups (parental race, language spoken etc.).

#### 2.1.3. Outcomes

The DMFT index is the predominant population-based measure of dental caries experience worldwide [28], providing the cumulative sum of an individual’s decayed, missing, and filled teeth [28]. While the index for the deciduous dentition is represented as dmft, the permanent dentition is identified as DMFT. We will consider all published and unpublished literature that has assessed the prevalence or severity of dental caries using the mean decayed (D), missing (M), and filled teeth (F) (DMFT/dmft) index. The primary outcomes will be: (1) caries severity, as assessed through the mean DMFT/dmft index, alongside its standard deviation (SD); and/or (2) caries prevalence (%), defined as the percentage of the population with a DT/dt higher than zero. The secondary outcomes will be the mean number of DT/dt, MT/mt, and FT/ft for each group expressed as the mean ± SD.

Additionally, we would classify the different methods and concepts used for identifying racial/ethnic groups and we would identify if the authors differentiated between these two terminologies. The presence of one or more of the following criteria will identify the failure to distinguish between both categories: (1) use of compound terms racial/ethnic or race/ethnicity; (2) use of terms “race” and “ethnicity” interchangeably; and (3) comparison of race and ethnic groups (e.g., comparing Whites to Mexicans).

#### 2.1.4. Types of Studies

An electronic search will be conducted of the relevant databases from their inception until December 2021. Cross-sectional studies, case–control studies, and cohort studies assessing the association between race and our primary outcomes will be included. Studies will not be excluded based on population sampling techniques (e.g., probabilistic or convenience sampling). Baseline data from experimental studies, such as randomized controlled trials and quasi-experimental studies, will be included if they address the research question and refer to a point in time before the intervention carried out in the original study.

Studies will be included according to the following criteria: (1) original research/secondary data focused on reporting the prevalence and/or severity of dental caries based on the WHO criteria of DMFT; (2) comparative studies having data on both racial/ethnic groups and nationally representative population; (3) data collected through dental examinations and clinical recording of either dental caries prevalence and/or the number of decayed, missing and filled teeth.

If more than one study used the same secondary dataset (e.g., national oral health surveys) and analyzed the same population more than once, the most recent analysis will be included. Studies that have a larger sample size and/or assess the sample over a longer period will be included over smaller studies in case they include the same population.

The exclusion criteria will be (1) studies that do not assess caries prevalence or severity; (2) absence of comparative majority population; (3) studies that use indices or assessment scores other than the DMFT (e.g., DMFS, which refers to decayed, missing, filled surfaces or SiC, significant caries index); (4) papers that define race based on immigration status; (5) studies that rely on self-reported clinical data or which lack clinical oral examinations for caries assessment; and (6) publications that refer to case reports/case series, conference abstracts, animal studies, in vitro studies, literature reviews, qualitative studies, letters, commentaries, opinion pieces, and editorials.

The language will not be a restriction for study inclusion. For articles published in languages other than English, a direct translation will be carried out. In case this is not possible, online translating tools will be used, such as Google Translate and FineReader (ABBYY FineReader PDF 15 for Windows).

#### 2.1.5. Context

The review will consider studies that are original, and performed in community settings, homes, mobile dental centers, schools, or dental hospitals. Any study that used data from national oral health surveys, government registries, or census will be included.

### 2.2. Information Retrieval

#### 2.2.1. Search Strategy and Information Sources

An initial electronic search was conducted in MEDLINE/PubMed to identify all potentially relevant articles on racial/ethnic inequities in dental caries from the inception of the platform until 31 December 2021. With the pilot search, all the relevant terms (i.e., controlled vocabulary terms and text words) were identified and the main search strategy was formed (Appendix B). This will be followed by a comprehensive and exhaustive search conducted in Scopus, Embase, Dentistry and Oral Sciences Database (EBSCOhost), the University of New Mexico Native Health Database, Bibliography of Native North Americans, and the Australian Indigenous HealthInfoNet. Keywords will include, “ethnic”, “race”, “minority,” and “dental caries”. A reference list with all selected articles will be compiled, and an additional freehand literature search of the bibliographic index of selected articles will also be conducted.

A systematic approach to source grey literature will be developed to identify all the relevant reports, by using Open Grey and ProQuest. Review authors will conduct an online web search using a combination of keywords, such as racial minority, ethnic group, Indigenous, and dental caries for the grey literature. Government reports, national oral health surveys, government registries that have comparative data on dental caries between race- or ethnicity-based groups will be considered for inclusion. All potentially eligible websites will be screened by two reviewers independently. Furthermore, expert authors in the field will be contacted via email for additional unpublished data or any other possible relevant studies.

#### 2.2.2. Study Selection Process

Studies retrieved from database searches will be uploaded to EndNote X9 (Clarivate Analytics, PA, USA). The final list of articles will be transported to Covidence, an online software platform, to assist with file management [29]. Two researchers (SN and SS) will individually screen all the titles and abstracts and assess the full texts of the selected studies against the inclusion criteria. All excluded studies will be listed, and a justification will be provided for their exclusion. Disagreements at any stage of the selection process will be resolved through discussion with a third reviewer (LMJ).

#### 2.2.3. Data Extraction

A customized data extraction form will be prepared, pilot-tested, and modified accordingly (Appendix C). Data extraction will be carried out by two calibrated independent reviewers (SN and SS). A third reviewer (LMJ) will be consulted for consensus if there are any discrepancies between the two reviewers. In case of missing data, the authors of the relevant papers will be contacted via email to obtain further information.

The data extraction form will include:Article information: Last name of the first author, year of publication, journal, and country or region of analysis;Study characteristics: Area and location of the study, sample size calculation (yes/no), study design, and sampling methods; details of the secondary dataset used (if relevant);Participant characteristics: Description of case population and control population, number of participants in each group, and mean age of participants;Defining ethnic/racial group: concept used to define a group (race, ethnicity, both, or neither), methods used for assessing race and ethnicity (self-reported, pre-existing records, defined by other, not stated), differentiation of both groups (yes/no), classification based on the taxonomies, any other variables used along with race and ethnicity, whether the authors differentiate race and ethnicity and whether they recognize the limitation of the classification, and stated purpose of use (cultural variable, demographic variable, sociodemographic, not stated);Dental caries measurement: Prevalence (%) of dental caries, mean DMFT/dmft, mean number of decayed teeth, missing teeth and filled teeth;Outcome: The results of selected study.

If the values are given in 95% confidence interval (CI), a formula recommended in the Cochrane Handbook, would be used for conversion to SD [30]. If 95% CI is missing for dental caries prevalence, it will be mathematically calculated based on the proportions [31]. Data extraction from graphs or charts will be created using WebPlotDigitizer tool (Version 4.2 GNU Affero General Public License).

### 2.3. Risk of Bias Assessment and Quality of Evidence Assessment

Quality assessment of included studies will be performed using the Newcastle–Ottawa Scale [32]. The scale uses three broad categories to judge each study: the selection of study groups, the comparability of the groups, and the ascertainment of the outcome of interest. The tool awards one star for each numbered item within the selection and outcome categories. A maximum of two stars can be given for comparability.

The quality of evidence across the included studies will be assessed using the Grading of Recommendations Assessment, Development and Evaluation (GRADE) methodology [33]. No study would be excluded based on the quality of the paper.

### 2.4. Data Synthesis

The results will be synthesized both narratively and quantitatively. A descriptive review of all included studies will be incorporated in a table of study characteristics. The data will be summarized on the different concepts/methods used for classifying race/ethnic groups.

#### 2.4.1. Meta-Analysis

For quantitative synthesis, meta-analysis will be conducted for mean DMFT scores, and dental caries prevalence based on each racial/ethnic category. A random-effects model using the restricted maximum likelihood (REML) method will be used in Stata statistical software (version 17.0, Stata Corporation, College Station, TX, USA) for meta-analysis. Forest plots will be constructed, and the effect size computed using Hedges’s g standardized mean and its corresponding 95% confidence interval (CI) [34]. The magnitude of the Hedges’s g will be interpreted as small (0.2), medium (0.5), or large (0.8). The random-effects meta-regression would be performed using the REML method. The goal of the meta-regression would be to explore and explain the between-study differences as a function of a moderator such as age, geographic location, and different classification of race/ethnicity. We would additionally calculate the power of random-effects meta-analyses and the average power of individual studies that contribute to the meta-analyses and compare the power in between studies [34].

#### 2.4.2. Heterogeneity

Heterogeneity will be assessed using several methods as listed below:Forest plots and I^2^ statistics would be inspected for heterogeneity. The I^2^ index will be interpreted as low, moderate, or high inconsistency if the values are equal to 25%, 50%, and 75%, respectively [35].Galbraith plots (scatterplots for detecting potential outliers around the regression line) [36].Leave one out meta-analysis that calculates the effect size multiple times by omitting one study at a time.Depending on the total number of studies, subgroup analysis may be conducted based on age, geographic location, study design, year of publication and quality assessment score.

#### 2.4.3. Publication Bias

Publication bias will be identified using Egger’s test and funnel plot asymmetry [37]. An asymmetrical funnel plot may suggest the presence of publication bias. Studies with smaller samples are likely to report larger effect sizes; in order to overcome this, a “trim-and fill” funnel plot will be utilized [38]. The trim and fill function is able to balance the asymmetrical plot by “filling” additional study points. This method for computing publication bias provides a more robust estimate of the altered effect size.

## 3. Discussion

This review aims to quantify the magnitude of racial–ethnic inequities in dental caries at a global level. Dental caries cause destruction of the dental hard tissue (enamel and dentin) as a result of acids produced from the microbial fermentation of free sugar. The global prevalence of dental caries ranks first among all diseases [39] and 2.5 billion people are affected by untreated dental caries worldwide [40]. Dental caries experience can be clinically evaluated by the DMFT index. This oral health index is an indicator of both inequitable past treatments (M and F) and recent inequities of untreated dental disease (D) [28]. Cultural disparities in addition to socio-economic positioning suggest considerable heterogeneity in the distribution and, the magnitude of dental caries among racial or ethnic minority populations in society [41].

Different classification systems can have a direct impact on clinical interpretation, diagnosis, and outcome, and lead to either under- or over-estimation of the dental caries inequities among a population. Every ethnic and racial classification system contains inaccuracies and is therefore inadequate to capture the full ethnic and racial diversity of a given population [42]. Many of these systems confuse, obscure, or are considered offensive [42]. There have been inconsistencies in the literature on how race, ethnic and cultural groups are defined, often leading to meaningless classifications. Being social constructs, the definitions themselves change over time with changing societal norms, understanding, and expectations. In the global context, oral health research has consistently used varied racial and ethnic categories, with many researchers failing to include race or ethnicity as a variable entirely [43].

The concept of inequity as used in the World Health Organization (WHO) refers to differences in health that are avoidable and unnecessary and considered unfair and unjust [44]. Therefore, the difference between inequality and “inequity” is related to the ethical dimension attributed to health differentials. Inequity in oral health simply refers to the unequal opportunity for members of less privileged social groups, such as poor people, racial, ethnic, or religious groups, women, and rural dwellers for achieving good oral health [44]. In operational terms, pursuing dental health equity means eliminating oral health disparities that are associated with social disadvantage and marginalization. In relation to equality in dental caries, the concept is predicated on the assumption that each citizen will equally benefit from receiving the same dental services to treat caries, or benefit from health policies to prevent dental caries (water fluoridation, sugar taxation, affordable access to bottled drinking water as opposed to sugar-sweetened beverages in areas of poor water quality) [45,46]. In contrast, the concept of equity is centered on providing dental services to those most disadvantaged in society (children, via the Child Dental Benefits Scheme in Australia and the state and territory-funded school dental system) or publicly funded dental services for adults on low-incomes or social welfare recipients [47]. Thus, equity, by its definition, will posit that those who require support will/should receive it whilst those with sufficient financial means and health literacy will not.

Racial and ethnic minoritized groups are disproportionately represented among the socially disadvantaged. Indeed, racial and ethnic inequities are often used as a proxy for the impact of social processes of racialization and experiences of racism [48]. These inequities arise from complex interactions at multiple levels in society and are affected by biological, cultural, and social factors [49]. Another important social determinant is the historical consequence of colonialism [50]. The colonial history of a country also tends to shape the structure of the health/welfare system, and so disentangling their individual contribution to shaping health inequalities both would be very challenging, but not impossible to consider. While we may not be able to consider this interaction in a formal analytical way, we will attempt to describe these inequalities according to different welfare/health systems structures, as well as colonial processes and this may potentially shed some light on the mechanisms that shape these oral health inequities. Although inequities in dental caries are only one of many adverse health outcomes experienced by racially and ethnically minoritized groups, it is preventable. Government-led initiatives that identify and deconstruct neoliberal political governance processes that overtly benefit the major population (typically White) at the expense of non-White groups is a key way in which many health inequities, oral health included, can be eradicated [51].

## 4. Conclusions

Quantifying dental caries inequities among racial–ethnic groups is the first step in planning and developing interventions targeted towards these populations. Culturally sensitive approaches are required to both improve population oral health overall, as well as to reduce oral health inequities. This review will also help us understand the current concepts of race and ethnicity being used in the cariology literature. As a dental research community, we should understand the ways of classifying race and ethnicity, and consider the ways we analyze and report the measures to understand the true impact.

## Data Availability

Not applicable.

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
