# Peer review of "A Global Perspective of Racial–Ethnic Inequities in Dental Caries: Protocol of Systematic Review"

_ijerph, 2022, doi:10.3390/ijerph19031390_

Round 1
Reviewer 1 Report
This manuscript provides a clear overview of how the authors will conduct a systematic review of an important topic regarding inequities associated with racial/ethnic variations at an international level. The overall search strategy, methodology for dealing with bias, duplication and resolution of disagreements follows standard and acceptable best practice for systematic reviews. The research questions are clear and the processes well defined.
However, 1. The outcome variables contain complex interactions between different national health systems, especially regarding access to dental services and funding elements of the health systems. How these interactions will be managed needs clarification. 2. Inequity itself has not been explicitly defined and needs clarification especially for readers who may not be familiar with the distinctions between inequities and inequalities. 3. In many nations, the complex interaction of colonial history and the evolution of health and welfare systems is often an important facet of inequity. It maybe useful for the authors to indicated how they may deal with this issue in their review? 4. Finally, the past two years of the COVID pandemic may have increased inequities influenced by race/ethic factors. It may be important to identify how such major disruptions to health and economic systems may influence the collection of appropriate published data and associations with the key research questions?
Author Response
Reviewer 1:
This manuscript provides a clear overview of how the authors will conduct a systematic review of an important topic regarding inequities associated with racial/ethnic variations at an international level. The overall search strategy, methodology for dealing with bias, duplication and resolution of disagreements follows standard and acceptable best practice for systematic reviews. The research questions are clear and the processes well defined.
However, 1. The outcome variables contain complex interactions between different national health systems, especially regarding access to dental services and funding elements of the health systems. How these interactions will be managed needs clarification.
Author’s reply: Our interpretation of these interactions relates to a lack or restricted access to quality dental services among specific population groups. Our systematic review does not aim to explain the reasons behind oral health inequities or inequalities. These interactions are not within the scope of the review, and as such, understanding how national health systems are at the core of oral health inequities is the topic for future work.
- Inequity itself has not been explicitly defined and needs clarification especially for readers who may not be familiar with the distinctions between inequities and inequalities.
Author’s reply: We agree with the reviewer and often both the terms can be confused and misinterpreted in oral health research. We have clarified both the terms in the discussion section from line 292-309.
“The concept of inequity as used in World Health Organization (WHO) refers to differences in health that are avoidable and unnecessary and considered unfair and unjust [44]. Therefore, the difference between inequality and "inequity" is related to the ethical dimension attributed to health differentials. Inequity in oral health simply refers to the unequal opportunity for members of less privileged social groups, such as poor people, racial, ethnic, or religious groups, women, and rural dwellers for achieving good oral health [44]. In operational terms, pursuing dental health equity means eliminating oral health disparities that are associated with social disadvantage and marginalization. In relation to equality in dental caries, the concept is predicated on the assumption that each citizen will equally benefit from receiving the same dental services to treat caries, or benefit from health policies to prevent dental caries (water fluoridation, sugar taxation, affordable access to bottled drinking water as opposed to sugar-sweetened beverages in areas of poor water quality) [45, 46]. In contrast, the concept of equity is centered on providing dental services to those most disadvantaged in society (children, via the Child Dental Benefits Scheme in Australia and the state and territory-funded school dental system) or publicly funded dental services for adults on low-incomes or social welfare recipients [47]. Thus, equity by its definition, will posit that those who require support will/should receive it whilst those with sufficient financial means and health literacy will not.”
- In many nations, the complex interaction of colonial history and the evolution of health and welfare systems is often an important facet of inequity. It may be useful for the authors to indicated how they may deal with this issue in their review?
Author’s reply: We think this is an excellent point and this would add another interesting dimension to the paper. The associations between colonial history, welfare systems, and oral health inequities will not be empirically examined in the review. Instead, we will elaborate on these relationships when interpreting and critically analyzing the results of our review, particularly in the Discussion section of the upcoming review paper.
We have addressed this issue in the discussion section from line 315-322.
“Another important social determinant factor is the historical consequence of colonialism [50]. The colonial history of a country also tends to shape the structure of the health/welfare system, and so disentangling their individual contribution to shape health inequalities both would be very challenging, but not impossible to consider. While we may not be able to consider this interaction in a formal analytical way, we will attempt to describe these inequalities according to different welfare/health systems structure, as well as colonial processes and this may potentially shed some light on the mechanisms that shape these oral health inequities.”
- Finally, the past two years of the COVID pandemic may have increased inequities influenced by race/ethic factors. It may be important to identify how such major disruptions to health and economic systems may influence the collection of appropriate published data and associations with the key research questions?
Author’s reply: We agree with the reviewer that the COVID-19 pandemic may have increased the inequities. We will identify all the literature that has been conducted after or during the pandemic (2020-2021) and compare the findings to 2 years before the pandemic. Additional sub-group meta-analysis will help identify the differences. Although this analysis would be restricted by the existing literature.
Reviewer 2 Report
This protocol is clearly described, well concieved, and will offer an imporant contribution to our undestanding of the social and structural determinants of oral health. As a reader I look forward to understanding more about the way that racial-ethic inequalities are presented in the literature.
Author Response
This protocol is clearly described, well concieved, and will offer an imporant contribution to our undestanding of the social and structural determinants of oral health. As a reader I look forward to understanding more about the way that racial-ethic inequalities are presented in the literature.
Author’s reply: We would like to thank the reviewers with the positive feedback.
Reviewer 3 Report
- Title
The title communicates distinctly what the manuscript is about, identifying the report as a protocol of a systematic review; no unnecessary description reported.
- Abstract
The abstract provides an explicit statement of the main objectives the review addresses; it specifies the inclusion and exclusion criteria for the review protocol and the information sources used to identify studies and the date when each will last be searched. The name of the registry and registration number are not reported in the Abstract, as stated by the Prisma-P checklist, but in “Materials and Methods” section (line 85), this should be corrected.
- Introduction
The introduction describes the rationale for the review protocol in the context of what is already known, nevertheless, the explicit statement of the research questions the review will address are not expressed in the introduction, as stated by the Prisma-P checklist, but in “Materials and Methods” section (lines 86-92).
- Materials and methods
The research questions (lines 86-92) should be relocated in the previous section “introduction”. The first research question proposed is “To quantify the magnitude of inequities in dental caries prevalence and severity (O) among racial-ethnic (E) groups (P) when compared to the non-minority
(C) in observational studies (S)?”; it should be revised the language and punctuation, what stand those letters for?
This section specifies the study characteristics to be used as criteria for eligibility for the review protocol. The study design and methods are appropriate for the research question and well detailed.
It is well explained how studies were selected according to eligibility and exclusion criteria. The information sources are well described with the planned date of coverage and the search strategy is correctly reported such that it could be repeated.The description of the mechanism that will be used to manage records and data is present. The process that will be used for selecting studies through each phase of the review and method of data extraction from reports are well explained in appropriate sections.
The prioritization of outcomes is present with rationale, but the numbering of primary and secondary outcomes should be reviewed. The paragraph should be better set and made more intuitive.
The anticipated methods for assessing the risk of bias of studies are correctly inserted in this section. The data synthesis is properly presented in this section, respecting the topics of the referring checklist. Also presented are the publication bias across studies and the quality of evidence assessment.
- Discussion and conclusion
In the discussion section, the future findings of the following review are logically explained by main topics, explaining the importance of quantifying dental caries inequities among racial-ethnic groups as the first step in planning and developing interventions targeted. The implications of the
findings for future research and potential applications are clearly considered.
Author Response
Title
The title communicates distinctly what the manuscript is about, identifying the report as a protocol of a systematic review; no unnecessary description reported.
- Abstract
The abstract provides an explicit statement of the main objectives the review addresses; it specifies the inclusion and exclusion criteria for the review protocol and the information sources used to identify studies and the date when each will last be searched. The name of the registry and registration number are not reported in the Abstract, as stated by the Prisma-P checklist, but in “Materials and Methods” section (line 85), this should be corrected.
Author’s reply: We agree with the reviewers and the PROSPERO registration number has been added in the abstract. Line 14-15.
- Introduction
The introduction describes the rationale for the review protocol in the context of what is already known, nevertheless, the explicit statement of the research questions the review will address are not expressed in the introduction, as stated by the Prisma-P checklist, but in “Materials and Methods” section (lines 86-92).
Author’s reply: We agree with the reviewers and the PICOS research question has been moved to the introduction. And we have defined the acronym for PECOS. Line 82-89
- Materials and methods
The research questions (lines 86-92) should be relocated in the previous section “introduction”. The first research question proposed is “To quantify the magnitude of inequities in dental caries prevalence and severity (O) among racial-ethnic (E) groups (P) when compared to the non-minority (C) in observational studies (S)?”; it should be revised the language and punctuation, what stand those letters for?
Author’s reply: The research question has been moved in the introduction section. The language and punctuation has been improved. Line 82-89.
This section specifies the study characteristics to be used as criteria for eligibility for the review protocol. The study design and methods are appropriate for the research question and well detailed.
It is well explained how studies were selected according to eligibility and exclusion criteria. The information sources are well described with the planned date of coverage and the search strategy is correctly reported such that it could be repeated. The description of the mechanism that will be used to manage records and data is present. The process that will be used for selecting studies through each phase of the review and method of data extraction from reports are well explained in appropriate sections.
The prioritization of outcomes is present with rationale, but the numbering of primary and secondary outcomes should be reviewed. The paragraph should be better set and made more intuitive.
Author’s reply: The numbering has been edited. The paragraph has been modified.
The anticipated methods for assessing the risk of bias of studies are correctly inserted in this section. The data synthesis is properly presented in this section, respecting the topics of the referring checklist. Also presented are the publication bias across studies and the quality of evidence assessment.
- Discussion and conclusion
In the discussion section, the future findings of the following review are logically explained by main topics, explaining the importance of quantifying dental caries inequities among racial-ethnic groups as the first step in planning and developing interventions targeted. The implications of the findings for future research and potential applications are clearly considered.